# NOTCH1- and CD117-Positive Stem Cells in Human Endometriosis and Adenomyosis Lesions

**DOI:** 10.3390/diagnostics14151642

**Published:** 2024-07-30

**Authors:** Dimitar Metodiev, Dimitar Parvanov, Margarita Ruseva, Rumiana Ganeva, Maria Handzhiyska, Nina Vidolova, Ani Chavoushian, Savina Hadjidekova, Georgi Stamenov

**Affiliations:** 1Department of Clinical Pathology, Nadezhda Women’s Health Hospital, 1373 Sofia, Bulgaria; 2Department of Research, Nadezhda Women’s Health Hospital, 1373 Sofia, Bulgaria; dimparvanov@abv.bg (D.P.); rum.ganeva@gmail.com (R.G.); mariavh@abv.bg (M.H.); nina.vidolova@abv.bg (N.V.); 3Department of Gastroenterology, Acibadem City Clinic UMBAL Mladost, 1784 Sofia, Bulgaria; 4Department of Medical Genetics, Medical University of Sofia, 1431 Sofia, Bulgaria; svhadjidekova@medfac.mu-sofia.bg; 5Department of Obstetrics and Gynecology, Nadezhda Women’s Health Hospital, 1373 Sofia, Bulgaria

**Keywords:** endometriosis, adenomyosis, stem cells

## Abstract

Adenomyosis and endometriosis are distinct gynecological disorders characterized by ectopic growth of endometrial tissue. Their etiology remains unclear, but stem cells have been implicated in both. The aim of this study was to investigate and compare the quantity of NOTCH1+ and CD117+ stem cells in endometriosis and adenomyosis lesions. Immunohistochemical staining of ectopic endometrium biopsies using antibodies against NOTCH1 and CD117 was performed. The quantity and spatial distribution of endometrial stromal cells positive for these markers were determined and compared between endometriosis and adenomyosis lesions. Additionally, their quantities were compared between endometriosis lesion types. Mann–Whitney U test showed that the median percentages of both NOTCH1+ and CD117+ cells in the endometriosis lesions were significantly higher than those in the adenomyosis lesions (2.26% vs. 0.13%, *p* = 0.002 and 0.44% vs. 0.26%, *p* = 0.016, respectively). Spearman’s test showed a positive correlation between NOTCH1+ and CD117+ cells in endometriosis lesions (R = 0.45, *p* = 0.027) but no significant correlation in adenomyosis lesions (R = −0.11, *p* = 0.69). The quantity of both stem cell types was highest in extragenital endometriotic lesions. Unlike adenomyosis, endometriosis lesions are associated with higher quantities of NOTCH1+ and CD117+ stem cells and a coordinated increase in their number. These findings support the distinct origin of the two conditions.

## 1. Introduction

Currently, endometriosis and adenomyosis are considered closely related to non-neoplastic gynecological conditions with similar pathophysiological expressions [1]. Both of them are characterized by ectopically located endometrium. Adenomyosis is defined as the presence of endometrial glands and stroma within the myometrium [2]. Endometriosis is defined as the presence of endometrial tissue outside the endometrium and myometrium [3]. Both conditions are estrogen-dependent, often present with similar symptoms and are associated with infertility. Endometriosis is estimated to affect around 10% of reproductive-age females globally [3], and adenomyosis presents in roughly 10% of women with subfertility, although reported prevalence varies widely depending on diagnostic method and location [4].

Stem cells are thought to play a significant role in the origin and development of endometriosis and adenomyosis [5,6,7]. Several versions of the stem cell hypothesis concerning the pathogenesis of endometriosis and adenomyosis exist [8,9]. It has been suggested that endometriotic and adenomyotic lesions could be monoclonal or polyclonal in origin [10,11]. Either a single or multiple stem cells can give rise to a lesion, and glands typically originate from a single stem cell [11]. Stem cells that express specific cell markers, such as NOTCH1 [12,13], CD117 [14], OCT-4 [15], Musashi-1 [16], SUSD2, N-cadherin or SSEA-1 [7], have been found in the eutopic endometrium of patients with endometriosis and adenomyosis. Recent studies revealed that CD117-positive stem cells are associated with estrogen and progesterone levels, hormone signaling and implantation success [17]. However, there are scarce data about the presence and quantity of specific stem cell subtypes in the endometriotic and adenomyotic lesions. The aim of this study was to compare the quantity of NOTCH1- and CD117-positive cells between endometriosis and adenomyosis lesions. A secondary objective was to assess the relationship between the quantities of these two stem cell types in both types of lesions.

## 2. Materials and Methods

### 2.1. Study Design and Participants

The present cohort study was conducted at Nadezhda Women’s Health Hospital, Sofia, Bulgaria, between November 2021 and September 2023 to identify and quantify stromal cells positive for two stem cell markers (NOTCH1 and CD117) in ectopic endometrial lesions. The main objective was to compare the percentage of NOTCH1- and CD117-positive cells between endometriosis and adenomyosis lesions. In addition, the association between NOTCH1- and CD117-positive cells in terms of quantities and spatial distribution was studied. Finally, the quantities of cells positive for the stem cell markers were compared between endometriosis lesion subtypes (Figure 1).

All study procedures were approved by the hospital’s Ethics Committee (project code 50/17 September 2021), and written informed consent was obtained from all participants.

### 2.2. Diagnosis of Endometriosis and Adenomyosis

Female patients suffering from dysmenorrhea, menorrhagia or pelvic pain seeking assisted reproductive therapy (intrauterine insemination or in vitro fertilization) were subjected to diagnostic laparoscopy and transvaginal ultrasound. Cases with identified endometriotic lesions or sonographic findings suggestive of adenomyosis underwent surgical procedures which were performed by specialized fertility surgeons. Clinical diagnosis was histopathologically confirmed by an experienced pathologist using surgically obtained biopsy material from affected tissue (containing ectopic endometrium) and appropriate staining techniques—hematoxylin–eosin (HE) and immunohistochemistry (IHC) markers for CD10 to visualize ectopic endometrial stroma and estrogen receptor (ER) to reveal ectopic endometrial glands. Endometriosis was classified according to the lesion location.

### 2.3. Immunohistochemistry Staining 

Lesion biopsy tissue was processed as previously described [18]. Briefly, 4 µm thick sections of affected tissue containing ectopic endometrium were cut on a manual microtome from formalin-fixed paraffin-embedded tissue blocks. Following deparaffinization in xylene and rehydration in graded ethanol solutions, the sections were rinsed with 0.01 M citrate buffer (pH 6), and 3% H_2_O_2_ was added to them to quench endogenous peroxidase activity. The next step was incubation with 0.4% casein in phosphate-buffered saline (PBS) to reduce non-specific binding of primary antibody, followed by 1 h incubation at 37 °C with rabbit polyclonal primary antibodies against NOTCH1 (E-AB-12815, Elabscience, Houston, TX, USA), CD117 (RB-9038-RQ, Epredia, Portsmouth, NH, USA), estrogen receptor (1-ES006-07, Quartett, Berlin, Germany) and mouse monoclonal antibody against CD10 (DAK-CD10 IR 786, Dako, Nowy Sącz, Poland). Following treatment with post-primary block (10% (*v*/*v*) animal serum in tris-buffered saline), poly-HRP anti-rabbit IgG was applied. Incubation with 3,3′-diaminobenzidine (DAB) chromogen was then performed to visualize the bound antibodies in the form of a brown precipitate. Ultimately, tissue was counterstained with 0.02% hematoxylin. Appropriate positive controls were used for each of the studied markers (human breast cancer tissue for NOTCH1, GIST for CD117, tonsil for CD10 and uterine cervix for ER). As a negative control, sections were treated with PBS instead of the respective primary antibody.

Additionally, specimens were stained with toluidine blue (05-M23001, Bio-Optica, Milan, Italy), which allows for visualization of acidic granules to ensure that CD117+ cells were not mast cells.

### 2.4. Microscopic Evaluation

Olympus CKX41 inverted microscope and Olympus EP50 wireless digital camera (Olympus, Melville, NY, USA) were used to capture high-resolution images of each stained ectopic endometrium sample at 40× magnification. The percentage of NOTCH1+/CD117+ cells (brown stain) and the number of endometrial stromal cells (blue counterstain) in the selected endometriosis and adenomyosis lesion compartments were determined using computer-assisted image analysis with HALO Image analysis platform (version 3.4, IndicaLabs, Inc., Corrales, NM, USA). The Spatial analysis module of the same software was used for measuring the mean distances between the two studied stem cell types. 

### 2.5. Statistical Analysis

Kolmogorov–Smirnov test was performed to check the distribution of the data. Continuous data are expressed as mean and standard deviation (SD) for normally distributed data; otherwise, median and range are reported. For each of the two stem cell markers (NOTCH1 and CD117), the percentages of positive cells were compared between endometriosis and adenomyosis lesions via Mann–Whitney U test. In addition, the distances between the two stem cell types were compared between endometriosis and adenomyosis lesions. Kruskal–Wallis test was utilized to compare the quantities of NOTCH1+ cells and CD117+ cells in endometriosis lesion subtypes. The relationship between the percentage of the studied two types of stem cells in the endometriosis lesions and adenomyosis lesions was analyzed with Spearman’s rank-order correlation coefficient (ρ). The significance level was set at *p* < 0.05. All data were analyzed using SPSS statistical software, version 21.0 (SPSS, Chicago, IL, USA).

## 3. Results

### 3.1. Study Population and Baseline Characteristics

Over the course of the present study, biopsies from endometriotic lesions (ovarian (*n* = 10), tubal (*n* = 4), pelvic peritoneal (*n* = 3), intestinal (*n* = 3) and cutaneous scar (*n* = 3)) were obtained from 23 female patients (Figure 2 and Figure 3), and myometrial biopsies (Figure 4) were obtained from 20 patients. Overall, the study population consisted of 43 women aged between 24 and 45 (mean of 37 years old). As summarized in Table 1, patients in the two study groups were comparable with respect to baseline characteristics, including age, body mass index (BMI), reproductive history and serum progesterone and estradiol level at the time of biopsy.

### 3.2. NOTCH1 and CD117 Expression in the Endometriosis and Adenomyosis Lesions

The initial toluidine blue staining revealed a negative reaction for granules, indicating that CD117+ cells were indeed stem/precursor cells and not mast cells.

The performed Mann–Whitney U test demonstrated that there were significantly higher quantities of both NOTCH1+ and CD117+ stem cells in endometriosis lesions (2.26% (0.08–37.11%) and 0.44% (0.11–15.45%), respectively) compared to adenomyosis lesions (0.13% (0.03–0.79%) and 0.26% (0.02–0.44%), respectively; *p* < 0.05) (Figure 5).

In terms of endometriosis lesion subtypes, the intestinal endometriosis lesions (Figure 3E,F) contained the largest quantities of both NOTCH1+ (11.5 ± 7.7%) and CD117+ endometrial stromal cells (7.9 ± 6.2%), followed by the cutaneous scar (11.4 ± 4.9% NOTCH1+ and 0.48 ± 0.35% CD117+ stromal cells). The lowest quantity of stem cells was observed in pelvic peritoneal endometriosis (0.46 ± 0.23% NOTCH1+ and 0.36 ± 0.11% CD117+ stromal cells).

### 3.3. Spatial Distribution of NOTCH1+ and CD117+ Cells in the Endometriosis and Adenomyosis Lesions

Spatial analysis revealed that the two stem cell populations were distinct, occasionally forming clusters in endometriosis lesions. Unlike NOTCH1+ cells, CD117+ cells were typically localized in close proximity to endometrial glands (Figure 6).

Proximity analysis showed that the mean distance between the two stem cell types was smaller in endometriosis lesions compared to adenomyosis lesions (83.5 ± 35.3 μm vs. 130.7 ÷ 43.7 μm, *p* < 0.05).

### 3.4. Relationship between NOTCH1- and CD117-Positive Stem Cells in Endometriosis and Adenomyosis Lesions

The performed Spearman’s correlation test showed that there was a positive correlation between the quantities of NOTCH1+ and CD117+ cells in endometriosis lesions (R = 0.45, *p* = 0.027). However, there was no significant correlation between these two stem cell phenotypes in adenomyosis lesions (R = −0.11, *p* = 0.69).

## 4. Discussion

Endometriosis and adenomyosis are disorders that involve ectopic growth of female genital tract tissue, which could lead to infertility and miscarriage [19]. Despite the existing perception of an association between adenomyosis and endometriosis, the two lesions likely arise through different pathogenetic mechanisms.

Cases with endometriosis of the pelvic peritoneum could be explained by the so-called retrograde menstruation hypothesis [20]. However, there are also reports of cerebral, pulmonary and other rare localizations in the literature. In one such extremely rare case, the authors associated the occurrence of cerebellar endometriosis with an available ventriculoperitoneal shunt due to hydrocephalus in a 40-year-old patient [21]. In other cases of cerebral endometriosis, the condition could be explained by circulating hematopoietic stem cells differentiating into endometrial glandular and stromal cellular elements [22,23,24]. Stem cell origin may also explain the isolated cases of endometriosis reported in male individuals in which the process develops against a background of hyperestrogenism (e.g., in cirrhotic patients) [25]. The results of the present study also favor a stem cell origin of endometriotic foci. Although there were no cases of cerebral or other rare localizations in our cohort, we observed the highest number of NOTCH1- and CD117-positive cells in cases with intestinal endometriosis and endometriosis at the site of the cutaneous scar after a Caesarean section, i.e., in so-called extragenital endometriosis. In contrast, we found the lowest number of cells with stem cell immunophenotype in patients with endometriosis of the pelvic peritoneum. 

The significantly lower numbers of NOTCH1- and CD117-positive cells in patients with adenomyosis compared with those with endometriosis point to the probability of a different pathogenetic mechanism of occurrence. Adenomyosis is commonly characterized by the presence of a non-functional basal endometrium and endometrial stroma within the myometrium [2]. Occasionally, ectopic endometrium shows phasic changes synchronous with those of the overlying functional endometrium. We observed such a finding in only one of our patients (Figure 4A). Another interesting observation in the same patient was the relatively higher number of cells with stem cell immunophenotype (Figure 4C,D). It is our opinion that in such cases, it is preferable to use the term myometrial endometriosis, which is distinct from conventional adenomyosis. The latter probably represents a kind of herniation of basal endometrium within the myometrium, and this process may be associated with chronic uterine autotraumatization with tissue damage and subsequent regeneration [26]. 

Quantifying and understanding the role of specific types of stem cells in these pathological conditions might provoke new treatment strategies and improved fertility management. Previous studies have shown that endometriosis is associated with a significant increase in the number of eutopic endometrial NOTCH1+ cells [12,13,27] and CD117+ cells [28,29,30] in comparison to a healthy control group. These changes might have an inhibitory effect on decidualization and lead to progesterone resistance and infertility, which are common features of endometriosis [31].

A potential limitation of the study was that adenomyosis patients were subjected to conservative surgery, whereby they underwent myometrial lesion resection instead of the gold standard for diagnosis—hysterectomy. This was performed as a fertility-sparing procedure that would allow them to undergo future assisted reproduction therapy. Nonetheless, this is the first study that characterizes and compares the quantity of ectopic endometrial stem cell populations between endometriosis and adenomyosis lesions.

In agreement with our findings that the number of both NOTCH1+ and CD117+ stem cells was markedly higher in the endometriosis lesions in comparison with adenomyosis lesions, recent evidence has found differential gene expression of downstream signaling targets of NOTCH1 and CD117 between deep infiltrating endometriosis and adenomyosis uteri. They reported that genes belonging to the PI3K pathway were increased in endometriosis lesions, and RAS pathway ones were increased in adenomyosis lesions [32]. CD117 is involved in the activation of both the PI3K pathway [33] and RAS pathway [34] and subsequent cellular survival and activation as a functional outcome [35]. In contrast, Notch1 pathway activation has been reported to inhibit PI3K/Akt/mTOR signaling, inducing apoptosis and autophagy in different tumors [36,37]. However, other evidence suggests Notch1 activation increases the expression of specific growth factor receptors, such as IGF1R and IL7Rα, and decreases p53 levels, indirectly leading to PI3K–AKT–mTOR1 activation [38]. 

Our study also demonstrated that the percentages of NOTCH1+ and CD117+ stem cells are strongly associated with the endometriosis lesions but not in the adenomyosis lesions. The observed higher quantities and synchronized protein expression in endometrial lesions were found also in other pathological conditions, such as in triple-negative breast cancer [39]. It might be a consequence of the positive regulation of Notch signaling by c-kit receptor tyrosinase kinase, as has been observed in epithelial neural stem cells [40] and progenitor T cells [41] by other authors. In addition, Notch signaling rapidly upregulates CD117 expression, which is important for T cell development and maturation [42,43], including TH2 cell differentiation [44]. It is possible that cooperation between CD117+ cells and Notch1+ cells contributes to the maintenance of stem cells in endometriosis lesions, similar to previous findings concerning neural stem cells [40]. In contrast, this regulation mechanism does not seem to be of major importance in adenomyosis lesions where Notch1-CD117 coordination does not exist.

Although endometriosis and adenomyosis are benign conditions, they are chronic and often debilitating, and any insight into their pathophysiological mechanisms may help improve diagnostic accuracy, aid the development of novel therapeutic strategies and ultimately enhance the quality of life of patients.

## 5. Conclusions

In summary, both endometriosis and adenomyosis lesions contain NOTCH1+ and CD117+ stem cells. However, the former is associated with higher quantities and a coordinated increase in the percentage of both stem cell types. This finding may well explain the appearance of endometriosis, particularly in cases with distant localizations. The results of the present study support the notion of distinct etiology of the two conditions. While adenomyosis could be the result of disruption of the boundary between the basal endometrium and myometrium due to autotraumatic effects, endometriosis, especially its extragenital forms, could be caused by the ectopic differentiation of pluripotent stem cells.

## Figures and Tables

**Figure 1 diagnostics-14-01642-f001:**
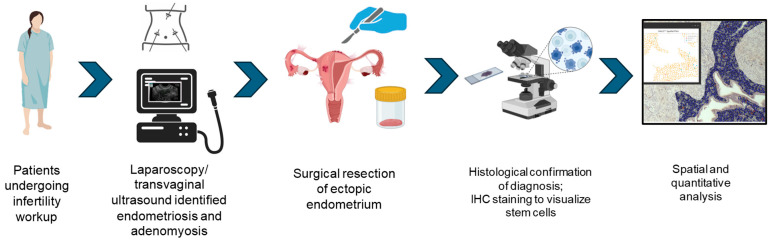
Study experimental design.

**Figure 2 diagnostics-14-01642-f002:**
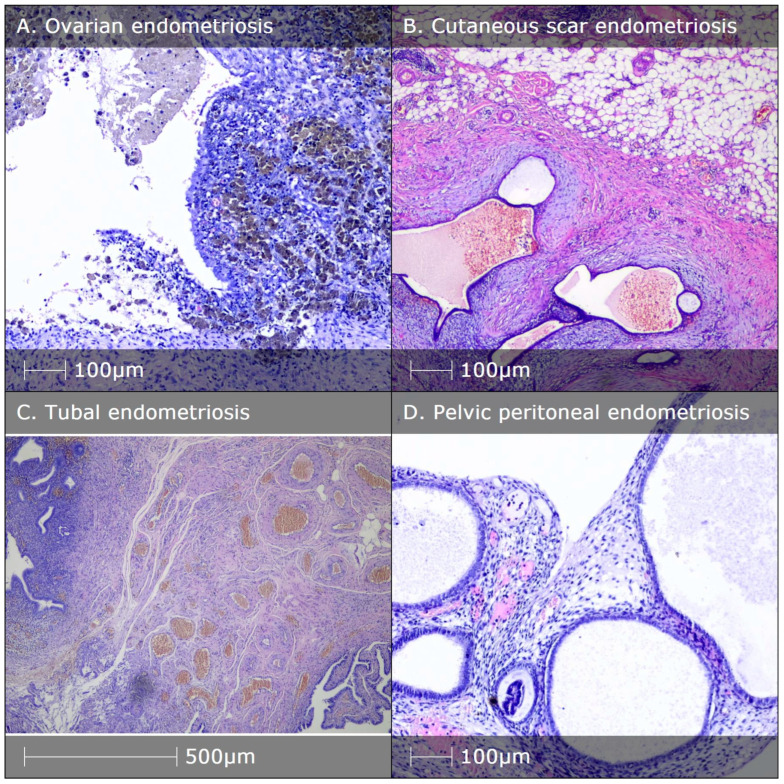
HE staining of different types of endometriosis lesions. (**A**) Endometriotic cyst filled with hemorrhagic material (so-called chocolate cyst); hemosiderophages in area of old hemorrhages; (**B**) soft tissues in area of cutaneous scar after Caesarean section with presence of cystic endometrial glands, surrounded by endometrial stroma; (**C**) tubal endometriosis in the upper left corner and tubal epithelium in the bottom right; (**D**) endometriosis of pelvic peritoneum with ectopic endometrial glands and stroma.

**Figure 3 diagnostics-14-01642-f003:**
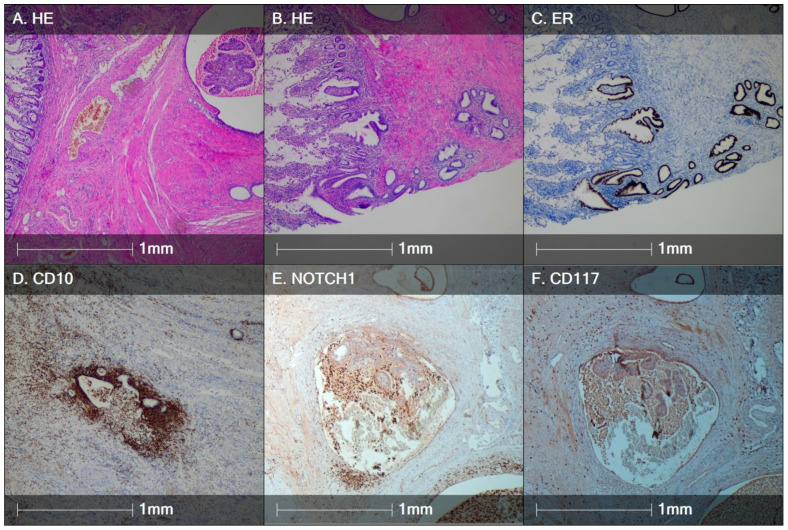
HE and IHC staining of a representative intestinal endometriosis lesion. (**A**) Intestinal endometriosis (within the ileum) with presence of cystic ectopic endometrial glands within the tunica muscularis propria; (**B**) intestinal endometriosis (purple) affecting the intestinal mucosa (left); (**C**) estrogen receptor expression in ectopic endometrial glands; (**D**) CD10 immunopositivity demarcating ectopic endometrial stroma; (**E**) NOTCH1+ cells (dark brown) in ectopic endometrial stroma (bottom) and intracystic debris; (**F**) CD117+ cells in ectopic endometrial stroma surrounding the ectopic cystic formation.

**Figure 4 diagnostics-14-01642-f004:**
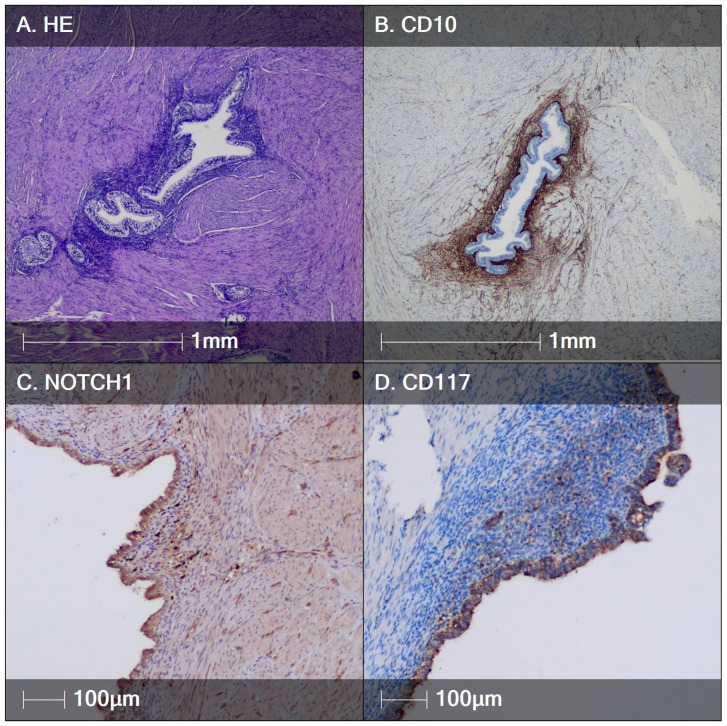
HE and IHC staining of an adenomyosis lesion with phasic changes. (**A**) Adenomyosis lesion with secretory changes* in the glands (apical and basal vacuoles); (**B**) CD10 marker visualizing endometrial stromal cells (dark brown stain) in the ectopic lesion; (**C**) NOTCH1+ cells within the ectopic endometrial stroma; (**D**) CD117+ cells within the ectopic endometrial stroma. * Phasic changes are a rare feature of adenomyosis. In such cases, it might be preferable to refer to the lesion as myometrial endometriosis.

**Figure 5 diagnostics-14-01642-f005:**
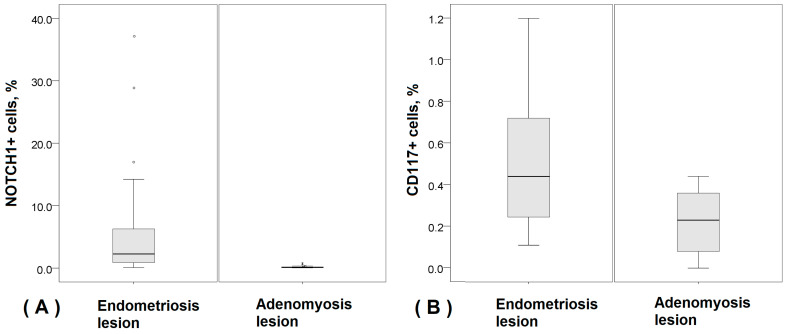
Box plots representing the percentage of studied stem cells in the stroma of patients with endometriosis and adenomyosis. (**A**) NOTCH1-positive cells in endometriosis (*n* = 23) and adenomyosis lesions (*n* = 20); (**B**) CD117-positive endometrial cells in endometriosis (*n* = 23) and adenomyosis lesions (*n* = 20).

**Figure 6 diagnostics-14-01642-f006:**
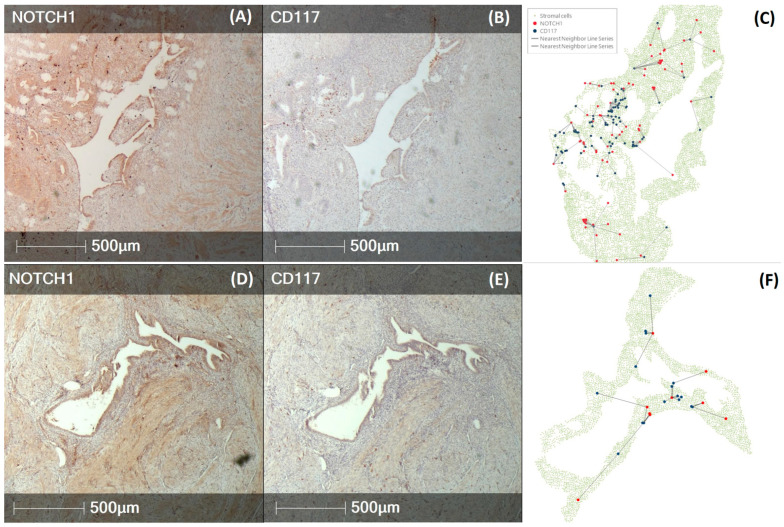
IHC showing NOTCH1 and CD117 expression in representative endometriosis and adenomyosis lesions. (**A**) NOTCH1-positive stromal cells (brown) in a representative tubal endometriosis lesion; (**B**) CD117-positive stromal cells (brown) in the same tubal endometriosis lesion; (**C**) spatial plot of the two types of stem cells in the same endometriotic lesion (NOTCH1+ cells—red, CD117+ cells—blue, ectopic endometrial stromal cells—green, lines—nearest neighbor between the two stem cell types); (**D**) NOTCH1-positive stromal cells (brown) in representative adenomyosis lesion; (**E**) CD117-positive stromal cells (brown) in the same adenomyosis lesion; (**F**) spatial plot of the two types of stem cells in the same adenomyosis lesion (NOTCH1+ cells—red, CD117+ cells—blue, ectopic endometrial stromal cells—green, lines—nearest neighbor between the two stem cell types).

**Table 1 diagnostics-14-01642-t001:** Baseline characteristics of studied patients with endometriosis and adenomyosis lesions.

Variables, Units	Endometriosis Lesions (*n* = 23)	Adenomyosis Lesions (*n* = 20)	*p* Value
Age, years	37.5 ± 4.5	36.8 ± 6.1	0.20
Body mass index, kg/m^2^	22.6 ± 3.3	23.1 ± 3.1	0.74
Endocrinology			
Serum estradiol, pg/mL	142.3 (41.5–321.4)	129.6 (22.6–365.1)	0.09
Serum progesterone, ng/mL	18.6 (6.2–63.0)	16.9 (7.8–65.9)	0.42
Reproductive history			
Prior ART attempts, *n*	0.4 ± 0.7	0.3 ± 0.5	0.17
Gravida	0.7 ± 0.5	0.6 ± 1.1	0.32
Para	0.2 ± 0.4	0.3 ± 0.2	0.63

Data presented as mean ± standard deviation or median and range in parenthesis depending on normality of distribution. ART—assisted reproduction therapy.

## Data Availability

The data presented in this study are available upon request from the corresponding author.

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
