# Peer review of "NOTCH1- and CD117-Positive Stem Cells in Human Endometriosis and Adenomyosis Lesions"

_diagnostics, 2024, doi:10.3390/diagnostics14151642_

Round 1

Reviewer 1 Report

Comments and Suggestions for Authors

This paper provides insight into the understanding of adenomyosis and endometriosis, which represent two significant gynecological disorders.

I have a few comments and suggestions.

Major objection:

There are numerous stem cell markers, including several that can be identified in the endometrium. The scientists' choice of markers for studying stem cells in adenomyosis and endometriosis remains unclear.

Minor objections:

1. "Either a single or multiple stem or stem cell-like cells might form a single lesion, with glands typically arising from a single cell".. lines 62,63....this sentence is quite unclear, I suggest simplifying it. 

2. "These changes might have an inhibitory effect on decidualization and could lead to progesterone resistance and infertility [31]"...lines 281,282...in this reference I did not find support for the statement before. In the reference [31] CD117 and NOTCH1 are not mentioned.

3. "In summary, both endometriosis and adenomyosis lesions contain NOTCH1+ and  CD117+ hematopoietic stem cells."...lines 321,322... hematopoietic stem cells are a subset of stem cells that specifically give rise to blood cells. Not all stem cells are hematopoietic stem cells.  So I think this is a mistake.

Author Response

Comment 1:  There are numerous stem cell markers, including several that can be identified in the endometrium. The scientists' choice of markers for studying stem cells in adenomyosis and endometriosis remains unclear.

Response 1: We would like to thank reviewer 1 for the time and thoroughness when reviewing our work. In response to the first comment, we agree that there are numerous markers used to identify stem cells in adult tissues. Nonetheless, according to the literature, most of these markers are identified and quantified using flow cytometry on a fresh tissue sample. Because of the nature of the study, the biopsy material was immediately fixed for histological examination and confirmation of the diagnosis by a pathologist. In addition, we wanted to explore the spatial distribution of the studied cells, which would not be possible using methods such as flow cytometry and gene expression studies. We chose these two markers specifically after trying multiple other immunohistochemistry markers reported in the endometrium including CD146, CD140b, CD34, SUSD2, most of which either revealed non-specific/diffuse staining or stained blood vessels and not the stroma. In addition, as mentioned in the discussion, other authors had reported differential expression of CD117 and NOTCH1 in the eutopic endometrium of endometriosis patients and healthy controls, so we were reassured these markers could be relevant to this condition.

Comment 2: 

  1. "Either a single or multiple stem or stem cell-like cells might form a single lesion, with glands typically arising from a single cell".. lines 62,63....this sentence is quite unclear, I suggest simplifying it.
  2. "These changes might have an inhibitory effect on decidualization and could lead to progesterone resistance and infertility [31]"...lines 281,282...in this reference I did not find support for the statement before. In the reference [31] CD117 and NOTCH1 are not mentioned.
  3. "In summary, both endometriosis and adenomyosis lesions contain NOTCH1+ and CD117+ hematopoietic stem cells."...lines 321,322... hematopoietic stem cells are a subset of stem cells that specifically give rise to blood cells. Not all stem cells are hematopoietic stem cells. So I think this is a mistake.

Response 2: To accommodate the reviewer’s minor objections, we have made the following modifications and clarifications in the text:

  1. "Either a single or multiple stem cells can give rise to a lesion, and glands typically originate from a single stem cell."
  2. "These changes might have an inhibitory effect on decidualization, could lead to progesterone resistance and infertility, which are common features of endometriosis [31]."

The sentence is a hypothesis of the consequence of the increased quantities of stem cells. The citation 31 refers to the common features of endometriosis.

  1. NOTCH1 and CD117 can give rise to blood lineages; however, we agree that this is irrelevant to our study and can cause confusion so we have removed the word “hematopoietic” from the text.

Reviewer 2 Report

Comments and Suggestions for Authors

This article contains a detailed study examining the various pathogenetic mechanisms and cellular characteristics of endometriosis and adenomyosis. The study focuses primarily on the role of NOTCH1+ and CD117+ hematopoietic stem cells in these diseases and provides insights into their importance for diagnosis and treatment. The text is well organized, with a logical flow that guides the reader through the research findings and involves comprehensive literature review. Although the fundamental basis and novelty of this study is controversial because there are a lot of similar investigations which result in nothing related to clinical implications. It brings a small piece for pathogenetic model but these results must be proved by experimental investigations because immunohistochemistry is only a top of the iceberg in endometriosis and adenomyosis pathogenetic model. The statistics paragraph should include the check fir the normality distribution and the tables should explain SD. Nevertheless the figures are brilliant.

Author Response

Comment 1:  This article contains a detailed study examining the various pathogenetic mechanisms and cellular characteristics of endometriosis and adenomyosis. The study focuses primarily on the role of NOTCH1+ and CD117+ hematopoietic stem cells in these diseases and provides insights into their importance for diagnosis and treatment. The text is well organized, with a logical flow that guides the reader through the research findings and involves comprehensive literature review. Although the fundamental basis and novelty of this study is controversial because there are a lot of similar investigations which result in nothing related to clinical implications. It brings a small piece for pathogenetic model but these results must be proved by experimental investigations because immunohistochemistry is only a top of the iceberg in endometriosis and adenomyosis pathogenetic model. The statistics paragraph should include the check fir the normality distribution and the tables should explain SD. Nevertheless the figures are brilliant.

Response 1:  

We appreciate the positive feedback of Reviewer 2.

To our knowledge, this is the first study that looks into the quantity and distribution of NOTCH1+ and CD117+ stem cells in the ectopic lesions of patients and compares the two conditions. Nonetheless, we agree that our study looks at a fundamental level and has no direct clinical implications.

We have added the relevant clarification regarding the test for normality used in the “Statistical analysis” subsection of the Materials and Methods (highlighted in yellow in the revised manuscript).